

# Does carrying heavy loads impact ground reaction forces and plantar pressures in intervention police officers?

Mario Kasović[1,2,*], Davor Rožac[3], Andro Štefan[1], Tomáš Vespalec[2] and Lovro Štefan[2,4,*]

[1] Department of General and Applied Kinesiology, Faculty of Kinesiology, University of Zagreb, Zagreb, Croatia
[2] Department of Physical Activities and Health Sciences, Faculty of Sports Studies, Masaryk University, Brno, Czech Republic
[3] Department of Kinesiology of Sport, Faculty of Kinesiology, University of Zagreb, Zagreb, Croatia
[4] Department of Physical Education, Catholic University in Ruzomberok, Ružomberok, Slovak Republic
[*] These authors contributed equally to this work.

## ABSTRACT

**Background**. The main purpose of the study was to examine whether heavier loads might have an effect on ground reaction forces and plantar pressures.

**Methods**. Ninety-six elite intervention police officers were recruited in this cross-sectional study. Ground reaction forces and plantar pressures beneath the different foot regions were evaluated using Zebris FDM pressure platform, while a graduate increase in load carriage was as following: (i) 'no load', (ii) 'a 5-kg load', (iii) 'a 25-kg load' and (iv) 'a 45-kg load'.

**Results**. Carrying heavier loads increased ground reaction forces beneath forefoot and hindfoot regions of both feet, and midfoot region for the right foot. For plantar pressures, increases beneath the hindfoot region of both feet and midfoot region of the right foot were observed, while carrying heavier loads.

**Conclusion**. This study shows significant increases in both ground reaction forces and plantar pressures, especially beneath the forefoot and hindfoot regions of both feet. Since the largest forces and pressures are produced beneath the hindfoot and forefoot, future research should pay special attention to these regions and their ground absorptions, additionally preventing from muscle and joint injuries.

## INTRODUCTION

Carrying excessive load represents a main component of personal mobility for successful competition of specific tasks (*Birrell, Hooper & Haslam, 2007*). To be able to perform at maximal level, special populations of military (*Knapik, Reynolds & Harman, 2004*; *Joseph et al., 2018*; *Walsh & Low, 2021*) and police (*Larsen, Tranberg & Ramstrand, 2016*; *Dempsey, Handcock & Rehrer, 2013*; *Lewinski et al., 2015*; *Ramstrand et al., 2016*; *Joseph et al., 2018*) personnel are required to execute highly demanding physical activities, including running, jumping and carrying heavy objects (*Lockie et al., 2019*; *Marins et al., 2020*). Although such

Corresponding author
Lovro Štefan,
lovro.stefan1510@gmail.com

equipment has protective effects for completing tasks and duties (*Walsh & Low, 2021*), evidence suggests that the load used often exceeds the recommended cut-off value of 45% body mass (*Andersen et al., 2016*; *Orr et al., 2015*). Thus, it is not surprising that extreme loading conditions may lead to changes in foot placement on the ground while absorbing various shocks during heavy load carriage (*Scott, Menz & Newcombe, 2007*; *Saltzman & Nawoczenski, 1995*). Thus, information on ground reaction forces and plantar pressures during load carriage may be relevant to describe the mechanisms of gait and to provide the magnitude of impact forces acting on the foot (*Birrell, Hooper & Haslam, 2007*). Moreover, both physiological and biomechanical costs of carrying heavy loads may alternatively lead to musculoskeletal and neurological injuries caused by greater forces being distributed on the foot (*Orr et al., 2015*; *Orr et al., 2021*). Indeed, a prolonged load carriage can lead to fatigue (*Fallowfield et al., 2012*), with longitudinal studies suggesting that knee, ankle and foot are the most common body sites of musculoskeletal pain (*Orr et al., 2015*; *Reynolds et al., 1999*).

Studying the effects of carrying heavy loads on ground reaction forces (*Goffar et al., 2013*; *Lenton et al., 2018*; *Majumdar et al., 2013*; *Sessoms et al., 2020*; *Tilbury-Davis & Hooper, 1999*; *Wang et al., 2023*; *Lenton et al., 2018*; *Dar et al., 2023*) and plantar pressures (*Goffar et al., 2013*; *Park et al., 2013*) has been mainly conducted among military personnel. Nevertheless, as one would expect, heavier loading conditions systematically lead to increases in both vertical and antero-posterior ground reaction forces produced during gait (*Walsh & Low, 2021*). Although the nature of an increase in ground reaction forces following heavier loads is somewhat expected, when the force is being observed on the surface as pressure, previous evidence has suggested that plantar pressures beneath different foot regions remain unchanged (*Goffar et al., 2013*). This would imply that force is simultaneously distributed under the specific foot regions and is not impacted by external load of different mass. Contrary to these findings, a recent study conducted among elite special police officers has shown significant changes in both ground reaction forces and plantar pressures beneath different foot regions while carrying heavy loads, pointing out that special population of police officers may be more prone to kinetic gait changes, compared to military active duty solders (*Kasović et al., 2023*). In specific, a study by *Kasović et al. (2023)* showed gradual increases in ground reaction forces and plantar pressures under forefoot, midfoot and hindfoot regions of both feet following heavier load carriage, while temporal gait parameters, including walking speed, remained unchanged (*Kasović et al., 2023*). This would imply that increases in force beneath both feet might be predominantly due to the static effect of the load rather than temporal changes of the system (*Birrell, Hooper & Haslam, 2007*). These findings are not in line with previous protective mechanisms of changes in ground reaction forces, where heavier loads increase double support or decrease walking speed (*Kinoshita, 1985*; *Birrell, Hooper & Haslam, 2007*; *Looney et al., 2021*). Some evidence has also suggested that the goal of loaded walking may even minimize upper body torque, leading to a reduced likelihood of injury (*LaFiandra et al., 2002*). Results from the kinematic data showed that the range of motion decreased in sagittal plane knee flexion and extension and pelvis rotation in the transverse plane, while increases in adduction/abduction and rotation of the hip were observed (*Birrell &*

*Haslam, 2009a*). Nevertheless, it has been confirmed that changes in ground reaction forces, especially in mediolateral direction are due to a decrease in stability during a single support gait cycle, shifting the body's center of mass further away from its neutral position (*Birrell, Hooper & Haslam, 2007*). Similar to special police officers, intervention police officers perform vigorous physical tasks and duties on a daily basis, accompanied by even heavier load carriage exceeding >50% of body mass, compared to military personnel (*Davis et al., 2016*; *Irving, Orr & Pope, 2019*). The examination of the effects of carrying heavy loads on gait kinetics would potentially lead for understanding the biomechanical responses of the gait which lead to an increased injury risk.

Therefore, the main purpose of the study was to investigate whether heavier loading conditions impacted ground reaction forces and plantar pressures of different foot regions in intervention police officers. We hypothesized, that heavier loads would gradually lead to increases in ground reaction forces beneath different foot regions, but limited evidence would be observed for increases in plantar pressures.

## MATERIAL AND METHODS

### Study participants

For the purpose of this study, data were collected as described in previous studies (*Kasović et al., 2023*; *Kasović et al., 2024*). Specifically, the sample size based on G\*Power calculation and using a standardized statistical power of 0.80, large effect size of 0.40 and $p < 0.05$ needed to be $N = 80$. However, we speculated that a certain drop-out rate might cause incomplete findings. By using a 20% enlargement, the final sample used for the analyses was $N = 96$. To be included in the study, participants needed to be a part of Intervention Police Unit for a minimum of three years and without acute or chronic diseases at the time of measurement. According to the Declaration of Helsinki (*World Medical Association, 2013*), all procedures performed in this study were anonymous and a written informed consent was signed by all participants. This study was approved by the Ethical Committee of the Faculty of Kinesiology and the Police Intervention Department under the Ministry of Internal Affairs of the Republic of Croatia (Ethical code: 511-01-128-23-1).

### Loading conditions

During testing, each participant walked over a platform and carried four types of standardized and prescribed loads proposed by the Ministry of Internal Affairs for intervention police officers: (1) body weight only ('no load'), (2) a 5-kg load ('load 1', a belt with a pistol loaded with a full handgun's magazine, an additional full handgun's magazine and handcuffs; mean weight for all participant ± SD =4.97 ± 0.25 kg), (3) a 25-kg load ('load 2', 'load 1' upgraded by a helmet, a ballistic vest and a multipurpose baton; mean weight for all participants ± SD = 20.02 ± 1.34 kg), and (4) a 45-kg load ('load 3', 'load 2' upgraded by additional protection for the lower extremities and a protective gas mask; mean weight for all participants ± SD = 45.10 ± 4.33 kg). The order of the other load was randomized, to reduce the impact of a learning effect (*Kasović et al., 2023*).

## Ground reaction forces and plantar pressures

Ground reaction parameters recorded from the software were maximal forces beneath the forefoot, midfoot and hindfoot regions of both feet (N). Plantar parameters included peak pressures beneath the same regions of both feet ($N/cm^2$). Of note, the software generated the zoning of both feet. For the dynamic measurements, the load distribution beneath the forefoot, midfoot and hindfoot regions of the feet is recorded during walking over the pressure platform. Assuming normal gait without deviations or acute/chronic conditions, the load distribution under the feet during gait is shown by a semispherical load distribution under the hindfoot, followed by a contact of the entire foot with the exception of the area of the medial longitudinal arch and an even load distribution under the forefoot (the maximum load during gait is often distributed under the big toe or under the center of forefoot). Although cut-off points for high pressure have yet to be established, according to Zebris manual (Zebris Medical GmbH), the maximum load should not exceed $40{\sim}N/cm^2$ under the heel and $55{\sim}N/cm^2$ under the forefoot and all the toes should support the force exerted on the foot.

## Testing procedure

To be able to calculate ground reaction forces and plantar pressures, we used a pedobarographic platform (ZEBRIS company, FDM; GmbH, Munich, Germany; number of sensors: 11,264; sampling rate: 100 Hz; sensor area: 149 cm × 54.2 cm), a simple and easy-to-administrate tool to investigate gait characteristics and followed the testing procedure in similar populations (*Kasović et al., 2023*; *Kasović et al., 2024*). Specifically, each participant walked barefoot over a platform for eight consecutive times at a self-selected walking speed with a different external load. Before and after the platform, two custom-made wooden platforms were placed, in order to establish normal gait. When the measurer gave the signal, the participant started to walk over the platform and when the end of a walkway was reached, the participant stopped, turned around and started walking towards the starting point. A cross-correlation analysis of all eight trials showed excellent reliability properties ($r > 0.90$). Once the measurement was completed, the load was removed and the participants were allowed to have a resting period for at least 3 min or when heart rate was below 100 beats per minute (*Seay et al., 2014*).

## Data analysis

The Kolmogorov–Smirnov test was used to assess the normality of the distribution. For normally distributed variables, basic descriptive statistics are presented as mean and standard deviation (SD). For not normally distributed variables, median and interquartile range (25th–75th) were applied. A one-way repeated measures ANOVA or the Friedman test were used to examine the differences between each loading condition. We used a Bonferroni *post-hoc* test to examine significant main effects. All statistical analyses were performed by using SPSS v23.0 software (IBM, Armonk, NY, USA) with an alpha level set a priori at $p < 0.05$ to denote statistical significance.

**Table 1  Changes in ground reaction forces and plantar pressures under the different loading conditions.**

| Study variables | 'No load' | 'Load 1' | 'Load 2' | 'Load 3' | Main effect | |
|---|---|---|---|---|---|---|
| **Ground reaction forces (max.)** | **Mean (SD)** | **Mean (SD)** | **Mean (SD)** | **Mean (SD)** | $F$($p$-value) | $\eta 2$ |
| Forefoot-L (N) | 852.3 (109.9)[b,c,d,e] | 873.0 (166.0) | 960.6 (115.1) | 978.4 (108.9) | 23.362 (<0.001) | 0.156 |
| Forefoot-R (N) | 865.6 (113.8)[b,c,d,e] | 893.0 (126.7) | 967.6 (115.2) | 984.4 (114.2) | 22.790 (<0.001) | 0.153 |
| Midfoot-L (N) | 170.6 (70.3) | 170.0 (74.2) | 187.8 (75.4) | 191.9 (82.6) | 2.178 (0.090) | 0.017 |
| Midfoot-R (N) | 173.9 (68.7)[c] | 178.1 (75.4) | 202.1 (81.4) | 206.6 (82.0) | 4.438 (0.004) | 0.034 |
| Hindfoot-L (N) | 588.6 (89.9)[b,c,d,e] | 609.5 (82.0) | 651.3 (86.6) | 662.2 (90.4) | 15.114 (<0.001) | 0.107 |
| Hindfoot-R (N) | 568.1 (84.5)[b,c,d,e] | 580.5 (94.0) | 617.4 (87.3) | 636.6 (95.1) | 11.915 (<0.001) | 0.086 |
| **Plantar pressures (max.)** | | | | | | |
| Forefoot-L (N/cm$^2$) | 43.8 (9.4) | 43.7 (8.5) | 45.7 (9.4) | 46.2 (9.7) | 1.843 (0.139) | 0.014 |
| Forefoot-R (N/cm$^2$) | 44.3 (9.7) | 45.2 (9.7) | 47.2 (10.6) | 47.7 (10.8) | 2.474 (0.061) | 0.019 |
| Midfoot-L (N/cm$^2$) | 16.6 (5.8) | 16.9 (6.4) | 18.4 (6.4) | 18.2 (6.5) | 2.220 (0.085) | 0.017 |
| Midfoot-R (N/cm$^2$) | 15.8 (5.4) | 16.3 (5.7) | 18.0 (6.2) | 18.2 (6.8) | 4.090 (0.007) | 0.031 |
| Hindfoot-L (N/cm$^2$) | 32.8 (6.9) | 33.2 (6.6) | 35.1 (7.0) | 35.7 (6.7) | 4.228 (0.006) | 0.032 |
| Hindfoot-R (N/cm$^2$) | 31.6 (7.0) | 32.1 (6.7) | 33.7 (6.8) | 34.5 (6.3) | 3.983 (0.008) | 0.031 |

Notes.
[a]Significant differences between 'no load' *vs.* 'load 1'; [b]significant differences between 'no load' *vs.* 'load 2'; [c]significant differences between 'no load' *vs.* 'load 3'; [d]significant differences between 'load 1' and 'load 2'; [e]significant differences between 'load 1' and 'load 3'; [f] significant differences between 'load 2' and 'load 3'.
$P < 0.05$

## RESULTS

Changes in ground reaction forces and plantar pressures underneath different foot regions are presented in Table 1. Carrying heavier loads led to significant increases in maximal ground reaction forces beneath the forefoot, midfoot and hindfoot regions of the foot. In specific, the largest magnitudes of changes were observed for left and right forefoot, followed by left and right hindfoot and right midfoot, while the area under the left midfoot did not show significant changes following heavier load carriage. Bonferroni *post-hoc* analyses showed significant differences between heavier load carriage, peak plantar pressures significantly increased for the right midfoot and right and left hindfoot regions, while forefoot regions of both feet and left midfoot did not significantly change. Although not the purpose of this study, we speculated that heavier loads might also impact walking speed: that is with an increased load the walking speed would gradually decrease. According to the data, walking speed remained statistically unchanged between the load conditions ('no load' = 4.44 ± 0.48 km/h; 'load 1' = 4.57 ± 0.53 km/h; 'load 2' = 4.59 ± 0.57 km/h and 'load 3' = 4.66 ± 0.68 km/h; $F$-value = 2.423, $p = 0.066$). Table 2 indicates the summary of the results in terms of an increase, decrease or no effect of load carriage on ground reaction forces and plantar pressures under the different foot regions.

## DISCUSSION

The main purpose of the study was to investigate whether heavier loading conditions impacted ground reaction forces and plantar pressures of different foot regions in intervention police officers. The findings suggest that: (i) carrying heavier loads increases ground reaction forces beneath forefoot and hindfoot regions of both feet, and midfoot

**Table 2** Summary of an increase, decrease or no effect of load carriage on ground reaction forces and plantar pressures for both feet.

| Foot regions | Significant main effects |
|---|---|
| **Right foot** | **Forces/pressures** |
| Forefoot | Increased/no effect |
| Midfoot | Increased/increased |
| Hindfoot | Increased/increased |
| **Left foot** | |
| Forefoot | Increased/no effect |
| Midfoot | No effect/no effect |
| Hindfoot | Increased/increased |

region for the right foot, and (ii) with heavier loads, plantar pressures beneath the hindfoot region of both feet and midfoot region of the right foot increase.

The results of this study are in line with previous findings conduced in military personnel (*Goffar et al., 2013*; *Lenton et al., 2018*; *Majumdar et al., 2013*; *Sessoms et al., 2020*; *Tilbury-Davis & Hooper, 1999*; *Wang et al., 2023*; *Kasović et al., 2023*; *Dar et al., 2023*). In a study by *Goffar et al. (2013)*, findings showed that carrying loads of 20 kg and 40 kg significantly increased ground reaction forces beneath all foot regions. The same study performed an interaction between load and arch (normal *vs.* low/high) and found significant main effects beneath medial forefoot, medial midfoot and lateral hindfoot. Unfortunately, the instrumentation used in this study was pre-programmed to generate the parameters beneath the three regions of the foot along the $y$ axis, while the information along the $x$ axis (medial/lateral direction) was not applicable. Another study conducted among 21 army reserve males found that tibiofemoral contact forces were greater while carrying loads of 15 kg and 30 kg, compared to unloaded condition (*Lenton et al., 2018*). In particular, the first peak of medial compartment contact force and second peak of total contact force increased in response to increasing load magnitude. Similar findings were observed in a study by *Majumdar et al. (2013)*, where added mass of >8.6 kg exhibited greater antero-posterior breaking forces and >6.8 kg greater antero-posterior propulsive forces, compared to unloaded condition. Moreover, a mass of >4 kg led to an increased peak vertical and propulsive impact forces, indicating that even smaller magnitudes of loads produced ground reaction force changes (*Majumdar et al., 2013*). Interestingly, a recent study by *Sessoms et al. (2020)* showed that only first (braking) and second (propulsive) peak of antero-posterior ground reaction forces changed with heavier loads, while no significant changes in vertical or medio-lateral ground reaction forces were observed. A study conducted in special police officers confirmed the findings of this study, where heavier loading conditions (5-kg, 25-kg and 45-kg loads) increased ground reaction forces beneath the forefoot, midfoot and hindfoot regions of both feet (*Kasović et al., 2023*). In general, a systematic review by *Walsh & Low (2021)* concluded that antero-posterior breaking and/or vertical peak forces gradually increased with heavier loads, while no changes in medio-lateral ground reaction forces were observed, which is often explained by

improvements in ergonomics and design in equipment over time and increases in power and work output during walking (*Tilbury-Davis & Hooper, 1999*).

Although evidence suggests that ground reaction forces increase during added mass (*Walsh & Low, 2021*), previous studies aiming to investigate the effects of carrying heavy loads on plantar pressure are inconclusive. For example, some studies reported increases in absolute plantar pressures (*Goffar et al., 2013*; *Park et al., 2013*; *Kasović et al., 2023*) and plantar areas (*Park et al., 2013*), while no effects for the relative distribution of plantar pressure on the plantar surface were observed (*Goffar et al., 2013*). The most recent study has shown gradual increases in plantar pressures beneath the forefoot, midfoot and hindfoot regions with heavier loads (*Kasović et al., 2023*). The results of this study indicated that the largest and significant changes were observed beneath the hindfoot region of both feet. The hindfoot region of the foot represents the first contact with the ground which closes a kinetic chain, absorbing vertical forces and stabilizing gait during heavy loads carriage (*Son, 2013*). This has been supported in previous studies, showing greater increases in peak plantar pressures beneath the medial and lateral hindfoot regions, compared to other regions of the foot (*Son, 2013*). Increases in plantar pressures while carrying heavy loads have been reported in previous systematic reviews (*Liew, Morris & Netto, 2016*; *Walsh & Low, 2021*) and explained by simultaneous increases in ground reaction forces exacerbated by greater breaking and propulsive forces (*Majumdar et al., 2013*; *Sessoms et al., 2020*; *Tilbury-Davis & Hooper, 1999*).

Increases in ground reaction forces (*Goffar et al., 2013*; *Lenton et al., 2018*; *Majumdar et al., 2013*; *Sessoms et al., 2020*; *Tilbury-Davis & Hooper, 1999*; *Wang et al., 2023*; *Kasović et al., 2023*; *Dar et al., 2023*; *Walsh & Low, 2021*) and plantar pressures (*Goffar et al., 2013*; *Park et al., 2013*; *Kasović et al., 2023*; *Walsh & Low, 2021*) following heavy loads carriage represent a natural response of the body to external mass, where excessive weight load increases muscular tension, particularly in lower extremities, producing larger forces and pressures in the forefoot and hindfoot regions. On the other hand, practical implications of this study may suggest that changes in ground reaction forces following heavier load carriage can lead to higher incidence of musculoskeletal injuries and disorders (*Orr et al., 2021*). Although we did not test the prevalence of body site injuries under different load conditions, previous studies have shown that lower back pain is the most prevalent body part being associated with prolonged heavy load (*Orr & Pope, 2016*), followed by knee, ankle and foot pain (*Orr et al., 2015*; *Reynolds et al., 1999*). When carrying heavy load, upper body forward lean is increased, stressing the vertebrae, intervertebral discs, muscles and spinal structures (*Orr et al., 2021*). Despite carrying heavy loads, acknowledging other associated factors with musculoskeletal pain, like walking/running volume (*Knapik, 2014*) special populations go through should be a cornerstone for implementing special policies and strategies for re-positioning load on the body and re-adjusting external mass. This is in line with previous findings, where constant load carriage over time may cause a sustained additional injury within the first 12 months of service, optimizing an injured soldier's rehabilitation process and returning to work (*Orr et al., 2017*). Also, by understanding mutual inter-correlations between external heavy loads, ground reaction forces and injuries and taking into account load mass, walking/running speed, distance covered, and

![PeerJ logo]

type of terrain, interventions aiming to enhance the level of physical conditioning during load carriage should be advocated.

This study has several limitations. First, we did not measure gait kinematics nor muscle activity properties during walking. Previous findings suggest that carrying heavy loads increases range of motions, joint impulses and moments and the activity of antigravity and propulsive trunk and leg muscles (*Walsh & Low, 2021*). Second, the participants were instructed to walk at self-selected speed, which can be a compensatory mechanism for altering gait locomotion to accommodate external heavy loads. By using a pre-determined treadmill walking speed, we might have observed different gait changes (*Birrell & Haslam, 2009b*). Alternatively, studies have shown that structured questionnaires aiming to assess subjective skeletal discomfort following a load carriage exercise of 1 h may be a practical tool for injury prediction (*Birrell & Haslam, 2009b*), which could have added more information about the musculoskeletal status of the participants in this study. Third, the load was not tested independently of how it was distributed on the body. Fourth, the testing procedure was based on walking barefoot, which is not a common practice during specific task performances. By using in-shoe insoles, we would be able to examine the effects in real situations, compared to laboratory testing. Finally, we observed somewhat asymmetrical changes between the left and the right foot, meaning that heavier loads did not impact both feet in the same magnitude. Although each participant was instructed not to target the pressure platform while walking towards it, it is possible that some participants were targeting pressure platform, unintentionally changing spatial and temporal patterns of the gait. Also, the asymmetry between the feet might have come from the first step being done with dominant *vs.* non-dominant foot and the compensatory mechanisms of force amortization when carrying heavier loads.

## CONCLUSION

In summary, carrying heavier loads has significant effects on ground reaction forces beneath the forefoot, midfoot and hindfoot regions and on plantar pressures beneath the hindfoot region in intervention police officers. Ground reaction forces and plantar pressures gradually increase with heavier loads, pointing out that it might be appropriate to consider the tradeoffs between necessary equipment, gait kinetics and risk of injury.

### Funding
The authors received no funding for this work.

### Competing Interests
The authors declare there are no competing interests.

### Author Contributions
- Mario Kasović conceived and designed the experiments, authored or reviewed drafts of the article, and approved the final draft.

- Davor Rožac performed the experiments, authored or reviewed drafts of the article, and approved the final draft.
- Andro Štefan performed the experiments, analyzed the data, authored or reviewed drafts of the article, and approved the final draft.
- Tomaš Vespalec analyzed the data, authored or reviewed drafts of the article, and approved the final draft.
- Lovro Štefan conceived and designed the experiments, performed the experiments, analyzed the data, prepared figures and/or tables, authored or reviewed drafts of the article, and approved the final draft.

## Human Ethics

The following information was supplied relating to ethical approvals (*i.e.*, approving body and any reference numbers):

the Ethical Committee of the Faculty of Kinesiology and the Police Intervention Department under the Ministry of Internal Affairs of the Republic of Croatia (Ethical code: 511-01-128-23-1).

## Ethics

The following information was supplied relating to ethical approvals (i.e., approving body and any reference numbers):

the Ethical Committee of the Faculty of Kinesiology and the Police Intervention Department under the Ministry of Internal Affairs of the Republic of Croatia (Ethical code: 511-01-128-23-1).

## Data Availability

The raw measurements are available in the Supplementary File.

## Supplemental Information

Supplemental information for this article can be found online at http://dx.doi.org/10.7717/peerj.16912#supplemental-information.

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
