# Peer review of "Does carrying heavy loads impact ground reaction forces and plantar pressures in intervention police officers?"

_PeerJ, doi:10.7717/peerj.16912_

## Round 0.1 · original submission · Major Revisions

The reviewers have suggested including additional literature (if relevant to your discussion) and linking the literature you cite to your findings.

There are also questions about the test protocols and data analyses that you should address in your rebuttal and revision.

Although the acceptance criteria for the journal do not include a subjective determination of the degree of advance or novelty, the manuscript should include a brief discussion of the relationship between the findings of this study and previous work perhaps highlighting some of the differences and similarities.

Reviewer 1 ·

Basic reporting

1. Some minor grammatical changes to be picked up in a proof read – e.g. line 74 “information of ground reaction forces”

Experimental design

2. Introduction - good rationale for study, but explanation of findings of Kasovic 2022 in brief would help to make the argument for why changes are expected here. For example – what are the proposed mechanisms for the changes in plantar pressure distribution and what task is this expected in (running, walking, what speeds etc).
3. Methods line 117-121 are these exact loads? It is surprising that the equipment comes exactly to 5 kg, 25 kg and 40 kg. It would also be good to understand if this was the officers own equipment or a set specifically for testing, and what protocol was followed to ensure PPE was well fitted and did not interfere with gait unduly. If this was the officers own pre fitted equipment, there will be some variance in the load reported and it would be good to report that here.
4. Testing procedure/Data analysis – description of statistical tests is thorough however, describing the variables entered into any analysis and how they were calculated or treated is appropriate prior to this section. This should include a list of variables such as peak planter pressures and the regions investigated. Information about how the zoning or masking of different foot regions was conducted and how data were treated such as filtering techniques and cut-off frequencies should also be detailed.
5. Testing procedures. Please detail how gait speed changed with load or how this was constrained. For example – participants walked at a preferred speed, but that speed will likely slow with the addition of load. This change in speed may affect the kinetic variables investigated and so should be measured or commented upon. Did you determine that the preferred speed of a participant was consistent through all the trials in a single condition? If so please detail how this was achieved, if not please detail how potential speed changes within conditions were accounted for within the analysis process. Alternatively, you may have allowed participants to pick a preferred speed initially and then constrained them to that speed in subsequent conditions and trials. Either way, some detail of this a reporting of the participants walking speeds is appropriate to help evaluate your findings.

Validity of the findings

6. Discussion, I am surprised by the asymmetrical findings between feet. Please could the authors comment on this. I would have expected any difference seen between conditions in the right foot to be replicated in the left foot – could there be some aspect of your experimental setup that explains this? For example could participants be targeting the plate and adjusting their gait for this differently depending on the foot they are landing with?
7. Lines 176-217. This literature is relevant but does not currently feel well linked to your findings. Perhaps some of this should be introduced in the introduction and used to develop your hypotheses and rationale whilst in your discussion you can use it to frame your findings in context.
8. General discussion. I believe there should be more focus on the relevance and implications of your results, for example are there specific injury implications from the changes you observed?

Additional comments

9. I believe some of your results would benefit from presentation in a figure. For example a figure of a foot pressure map, showing which sections of the foot changed pressure between conditions.

Reviewer 2 ·

Basic reporting

The study is fairly basic in nature, which does mean that the results are clear to understand, but I can't see it adding too much to the literature.
The study is very similar to a previous one the authors have published, I would say too similar at the moment without changes.
Some fundamental and seminal literature is missing, e.g. Kinoshita, H., 1985. Effects of different loads and carrying systems on selected biomechanical parameters describing walking gait. Ergonomics, 28, 1347–1362, LaFiandra, M., et al., 2002. Transverse plane kinetics during treadmill walking with and without load. Clinical Biomechanics, 17, 116–122, and Birrell, Stewart A. and Haslam, Roger A.The effect of military load carriage on 3-D lower limb kinematics and spatiotemporal parameters, Ergonomics, 52: 10, 1298-1304.

Experimental design

The design flaws are significant and highlighted in the limitations: Barefoot, uncontrolled walking speeds, and no gender comparisons which have a significant effect on walking patterns and GRF (see Subjective skeletal discomfort measured using a comfort questionnaire following a load carriage exercise, Birrell, S. A. & Haslam, R. A., 1 Feb 2009, In: Military Medicine. 174, 2, p. 177–182).

Validity of the findings

Basic statistics completed, results too similar to other by the authors, too many methodological differences to compare current results to the literature (uncontrolled speed, barefoot etc,).
We knew the conclusions to the paper 20 years ago (loads increase GRF), but this paper doesn't offer anything new as to why.

---

## Round 0.2 · accepted · Accept

I should request that you correct the text for minor language issues. Please change "In specific" to "Specifically" in lines 100, 228, and 262.

Reviewer 2 ·

Basic reporting

The paper is much improved with the clarity that has been added to the introduction and methodology. The results are well presented and easy to follow. The paper is well structured with clear conclusions. I think the authors did a good job amending the paper according to the reviewers comments.

Experimental design

Good sample size, and methodology is much better defined. RQs are fine but a little basic.

Validity of the findings

Findings are valid and reproducibly, stats look OK to me. My questions remain with the methodology and therefore transferability of the results (i.e. uncontrolled walking speed and no adjustment for walking speed within the GRF data, and walking in barefoot limit the use for the wider literature and comparison to what is already out there.

Additional comments

I will say that the authors have amended the paper very well, and taken on board mine and the other reviewers comments. I feel the paper is perfectly well written, with a good sample size, just with limitations that limit the transferability and scientific interest in the results..